# Insights into Regulation of C_2_ and C_4_ Photosynthesis in *Amaranthaceae*/*Chenopodiaceae* Using RNA-Seq

**DOI:** 10.3390/ijms222212120

**Published:** 2021-11-09

**Authors:** Christian Siadjeu, Maximilian Lauterbach, Gudrun Kadereit

**Affiliations:** 1Systematics, Biodiversity and Evolution of Plants, Ludwig Maximilian University Munich, 80638 Munich, Germany; g.kadereit@biologie.uni-muenchen.de; 2Martinstraße 20, 55116 Mainz, Germany; maxlauterbach0@gmail.com

**Keywords:** *Amaranthaceae*, C_4_ photosynthesis, Caryophyllales, *Chenopodiaceae*, complex trait evolution, gene regulation, *Salsola*, transcription factor

## Abstract

*Amaranthaceae* (incl. *Chenopodiaceae*) shows an immense diversity of C_4_ syndromes. More than 15 independent origins of C_4_ photosynthesis, and the largest number of C_4_ species in eudicots signify the importance of this angiosperm lineage in C_4_ evolution. Here, we conduct RNA-Seq followed by comparative transcriptome analysis of three species from *Camphorosmeae* representing related clades with different photosynthetic types: *Threlkeldia diffusa* (C_3_), *Sedobassia sedoides* (C_2_), and *Bassia prostrata* (C_4_). Results show that *B. prostrata* belongs to the NADP-ME type and core genes encoding for C_4_ cycle are significantly upregulated when compared with *Sed. sedoides* and *T. diffusa*. *Sedobassia sedoides* and *B. prostrata* share a number of upregulated C_4_-related genes; however, two C_4_ transporters (DIT and TPT) are found significantly upregulated only in *Sed. sedoides*. Combined analysis of transcription factors (TFs) of the closely related lineages (*Camphorosmeae* and *Salsoleae*) revealed that no C_3_-specific TFs are higher in C_2_ species compared with C_4_ species; instead, the C_2_ species show their own set of upregulated TFs. Taken together, our study indicates that the hypothesis of the C_2_ photosynthesis as a proxy towards C_4_ photosynthesis is questionable in *Sed. sedoides* and more in favour of an independent evolutionary stable state.

## 1. Introduction

C_4_ photosynthesis is a carbon-concentration mechanism, enhancing CO_2_ at the site of ribulose-1,5-bisphosphate carboxylase/oxygenase (RuBisCO). This mechanism leads to a decrease in the oxygenation reaction of RuBisCO, which in turn decreases photorespiration, because fewer toxic compounds resulting from the RuBisCO oxygenation reaction need to be recycled [1]. C_4_ photosynthesis requires a series of biochemical, anatomical, and gene regulation changes compared with the C_3_ photosynthesis ancestor [2,3]. C_4_ photosynthesis has been a major subject in life science. Since the discovery of C_4_ photosynthesis more than 50 years ago, its evolution is still under debate [4]. The current model of C_4_ evolution relies heavily on the C_3_–C_4_ intermediate (including C_2_) photosynthetic types as evolutionary stepping stones towards C_4_ photosynthesis [5,6,7]. Most C_3_–C_4_ intermediate species utilise C_2_ photosynthesis, where a photorespiratory glycine shuttle and its decarboxylation by glycine decarboxylase (GDC) concentrate CO_2_ in a bundle sheath-like compartment [5]. The establishment of this glycine-based CO_2_ pump and the restriction of GDC activity in the bundle sheath cells (BSCs) is considered an important intermediate step in the evolution towards C_4_ photosynthesis [8]. However, the absence of C_4_ relatives in lineages with C_3_–C_4_ intermediate phenotypes indicates that C_2_ photosynthesis can be an evolutionarily stable state in their own right [9,10]. On the other hand, the hybrid origin of C_2_ photosynthesis has been suggested because the anatomy of hybrids obtained from artificial crosses of *Atriplex prostrata* (C_3_) and *A. rosea* (C_4_) resemble the C_3_–C_4_ intermediate using a glycine shuttle to concentrate CO_2_ [11].

Despite its complexity, the C_4_ pathway independently evolved in at least 61 lineages of both monocot and eudicot lineages [12]. In eudicots, the *Amaranthaceae*/*Chenopodiaceae* alliance has the largest diversity of C_4_ syndromes with 15 independent origins of C_4_ identified, ten of which belong to *Chenopodiaceae sensu stricto* [12,13,14,15,16,17,18]. The closely related lineages *Camphorosmeae* and *Salsoleae*, belonging to the goosefoot family (*Chenopodiaceae*), are rich in C_4_ phenotypes [18] and contain a number of C_3_–C_4_ intermediate species, including C_2_, proto-Kranz type and C_4_-like type species. Both lineages are found in steppes, semi-deserts, salt marshes, and ruderal sites of Eurasia, South Africa, North America, and Australia [18,19]. *Camphorosmeae* comprise subshrubs and annuals, predominantly with moderately to strongly succulent leaves with a central aqueous tissue [16]. Evolutionary radiation was later in *Camphorosmeae* (early Miocene) than in *Salsoleae* (early to middle Oligocene) [18]. In *Camphorosmeae*, C_4_ photosynthesis likely evolved two times in the Miocene and different photosynthetic types are recognised on the basis of leaf anatomy with several C_4_ phenotypes [18,20]. C_4_ photosynthesis in *Salsoleae* likely evolved multiple times and most species are C_4_ plants with terete leaves and *Salsoloid* Kranz anatomy in which a continuous dual layer of chlorenchyma cells encloses the vascular and water-storage tissue [18,21,22]. Therefore, these two sister groups constitute a central component allowing the investigation between and within each plant group for understanding the origin of C_2_ photosynthesis, the evolution of C_4_ photosynthesis and adjustments in gene regulation leading to different photosynthetic types. Indeed, new insight into C_4_ evolution were gained from studying *Salsoleae* lineage using high-throughput sequencing methods. A photosynthetic transition from C_3_ pathway in cotyledons to C_4_ pathway in leaves of the *Salsoleae* lineage (*Chenopodiaceae*, *Salsola soda* L.) has been identified [23]. This C_3_-to-C_4_ transition is thought to be a rather exceptional phenomenon since species conducting C_4_ pathway in all photosynthetically active tissues/organs are supposed to be the most abundant within this group. In addition, comparative transcriptomics revealed two proposed transporters associated with C_2_ and C_4_ photosynthesis [24]. However, for the *Camphorosmeae* lineage, transcriptome analysis and gene expression profiles of different photosynthesis types are still lacking. Moreover, for both lineages, differential gene expression of regulatory genes (e.g., transcription factors, TFs) involved in different photosynthetic pathways remains poorly understood.

The development of complex traits is controlled by the coordination of expression of many TFs and signaling pathways [25]. Thus, TFs play an important role in regulation of gene expression and are certainly responsible for the fine-tuning of the cell-specific expression patterns in C_4_ photosynthesis [26]. The characteristic expression pattern of PHOSPHOENOLPYRUVATE CARBOXYLASE (PEPC) in C_4_ plants (i.e., high abundance in mesophyll M and low abundance in BS cells), for example, could be controlled by a number of TFs from the ZINC FINGER HOMEODOMAIN (zf-HD) TF family [27]. Therefore, TFs are hot candidates for a stepwise evolutionary change of complex traits such as C_4_ photosynthesis. Reviewing nine studies on potential regulators of C_4_ photosynthesis in maize, Huang and Brutnell [28] found no TF consistently identified across these studies and suggested that consistent differential expression obtained between C_3_ and C_4_ sister lineages could be a more effective way to prioritise candidate TFs.

To fill this knowledge gap with new pieces of the puzzle, we (1) report transcriptome de novo assemblies and differential expression analysis between C_2_, C_3_, and C_4_ species of *Camphorosmeae* (*Amaranthaceae*) using RNA-Seq, and (2) assess transcriptional regulator elements involved in C_3_, C_2_, and C_4_ photosynthesis in the *Amaranthaceae*/*Chenopodiaceae* complex. In this latter objective, we merged transcriptome data generated in this study with the publicly available transcriptome data of C_2_, C_3_, and C_4_ species of the sister lineage *Salsoleae*.

## 2. Results

### 2.1. Descriptive Statistics of RNA Data and De Novo Assembly

Between 27.4 and 37.9 million reads remained after quality filtering (98.68–99.06%) and were de novo assembled for each of the three species (Appendix A). Reduction in contigs by clustering resulted in 26,842 (*Sed. sedoides*, C_2_), 33,1653 (*T. diffusa*, C_3_), and 34,278 (*B. prostrata*, C_4_) contigs with an open reading frame. The number of BUSCOs genes recovered was 88.1, 88.6 and 90.9% for *T. diffusa*, *B. prostrata*, and *Sed. sedoides*, respectively (Appendix A).

### 2.2. Differential Expression Genes (DEGs) within Camphorosmeae

In total, 10,513 transcripts were expressed in at least six of the eight species and downstream analyses focused on this dataset (Appendix A). Principal component analysis of log_2_ transformed read counts (normalised to TPM) showed that replicates of each species in *Camphorosmeae* were very similar, whereas different species were clearly distinct from each other (Figure 1). The first principal component explained 51.47% of the total variation and *Sed. sedoides* (C_2_) was positioned somewhere in between *B. prostrata* (C_4_) and *T. diffusa* (C_3_). This result was similar to what was found in *Salsoleae* [24] as the C_3_–C_4_ intermediate species was positioned in between the C_3_ and the C_4_ species. The second component, explaining 42.51% of total variation, separated *Sed. sedoides* from the other two species.

### 2.3. Functional Classification and Enrichment of DEGs within the Camphorosmeae

In all three species, transcriptional investment, defined as percentage of all read counts of transcripts (normalised to TPM) belonging to a particular MapMan category, was highest in the MapMan category ‘Not assigned’ (23.10–27.54%) (Figure 2, Appendix A). MapMan category ‘Photosynthesis’ was second highest in all three species; however, the amount differed between species from 11.24% in *B. prostrata* (C_4_), to 15.03% in *T. diffusa* (C_3_), up to 21.58% in *Sed. sedoides* (C_2_) (Figure 2, Appendix A). In general, high transcriptional investment in the category ‘Photosynthesis’ in *Sed. sedoides* (C_2_) was caused by higher transcription of many genes of the sub-category ‘Calvin cycle’. However, transcription of a gene encoding RUBISCO ACTIVASE in Arabidopsis (AT2G39730) was the main driver of the difference among species (*Sed. sedoides* compared with *T. diffusa*: log_2_ fold change of 2.09; *Sed. sedoides* compared with *B. prostrata*: log_2_ fold change of 3.72; Appendix A). Transcripts belonging to the categories ‘Protein biosynthesis’ (5.93–8.27%), ‘Protein degradation’ (4.45–5.98%), and ‘Protein modification’ (3.03–4.09%) were also highly abundant in all three species. In *B. prostrata*, the category ‘C_4′_ was higher (transcriptional investment of 4.54%) compared with *Sed. sedoides* (1.09%) and *T. diffusa* (0.58%; Figure 2, Appendix A). ‘Photorespiration’ was about twice as high in *Sed. sedoides* and *T. diffusa* (2.59% and 2.21%, respectively) compared with *B. prostrata* (1.14%). This difference was caused by higher transcription of most genes of the category ‘photorespiration’ rather than a few genes (Appendix A). The categories with the lowest transcriptional investment in all three species were ‘DNA damage response’ (0.10–0.14%), ‘Polyamine metabolism’ (0.16–0.29%), and ‘Multi-process regulation’ (0.18–0.34%).

### 2.4. Differential Expression of C_4_-Related Genes in C_3_, C_2_ and C_4_ Camphorosmeae Species

As observed in other C_4_ species, most genes encoding for proteins involved in C_4_ photosynthesis were significantly upregulated in the C_4_ species *B. prostrata* (C_4_) when compared with *Sed. sedoides* (C_2_) and *T. diffusa* (C_3_) (Table 1, Appendix A). Out of 18 C_4_-related transcripts, 16 and 13 transcripts encoding known C_4_ cycle proteins were significantly upregulated (*p* < 0.001) in *B. prostrata* (C_4_) compared with *T. diffusa* (C_3_) and *Sed. sedoides* (C_2_), respectively. ALANINE AMINOTRANSFERASE (AlaAT, Log_2_FC = 5.17) was the most abundant followed by PYRUVATE ORTHOPHOSPHATE DIKINASE (PPdK, Log_2_FC = 4.63), PHOSPHOENOLPYRUVATE CARBOXYLASE (PEPC, Log_2_FC =4.12), BILE ACID:SODIUM SYMPORTER FAMILY PROTEIN 2 (BASS2, Log_2_FC = 3.97), NADP-malic enzyme (NADP-ME Log_2_FC = 3.86), PEP/phosphate translocator (PPT, Log_2_FC = 3.61) in *B. prostrata* (C_4_) as compared with *T. diffusa* (C_3_). Conversely, BASS2 Log_2_FC = 3.51, PPdK Log_2_FC = 3.37, PHOSPHATE TRANSPORTER 1 (PHT1, Log_2_FC = 3.16), PEPC (Log_2_FC = 2.95), and AlaAT (Log_2_FC = 2.61) were in the top five of highly upregulated genes in *B. prostrata* (C_4_) when compared with *Sed. sedoides* (C_2_). However, not all C_4_-related geneswere significantly upregulated in *B. prostrata* (C_4_) as compared with *Sed. sedoides* (C_2_). TRIOSE PHOSPHATE TRANSLOCATORransporters [TPT (Bv8_194450_rkme.t1), a chloroplast dicarboxylate transporter isoform [DIT (Bv4_072630_xjai.t1)] and a CARBONIC ANHYDRASE isoform [CA (Bv8_194450_rkme.t1)] were significantly upregulated in *Sed. sedoides* (C_2_) when compared with *B. prostrata* (C_4_) (Appendix A). Interesting, PHT1 (Bv3_049110_qgnh.t1) was significantly upregulated in *T. diffusa* (C_3_) as compared with *Sed. sedoides* (C_2_) (Appendix A).

Twelve of the C_4_-related enzymes except ASPARAGINE SYNTHETASE (Asn Synthetase), SODIUM:HYDROGEN ANTIPORT (NHD) and a CARBONIC ANHYDRASE (CA) isoform (Bv6_148840_uffy.t1) were significantly upregulated in *Sed. sedoides* (C_2_) compared with *T. diffusa* (C_3_). These enzymes include typical C_4_ enzymes such as PEPC, NADP-ME, PPdK, PHT4, Ala-AT, and ASPARTATE AMINOTRANSFERASE (Asp-AT), as well as C_4_-associated transporters such as BASS2 and DIT.

### 2.5. Differential Expression of Key Photorespiration Genes in C_3_, C_2_ and C_4_ Camphorosmeae Species

Out of 14 transcripts associated with photorespiratory enzymes, 12 were annotated and assigned (Table 2). All 12 photorespiratory transcripts were significantly upregulated in *Sed. sedoides* (C_2_) as compared with *B. prostrata* (C_4_), including the core photorespiratory enzymes GLYCINE DECARBOXYLASE (GDC, T-, H-, P-, L-), GLUTAMATE:GLYOXYLATE AMINOTRANSFERASE (GGT) and SERINE HYDROXYMETHYLTRANSFERASE (SHMT). In *Sed. sedoides* (C_2_), GGT, two SHMTs, GDC-T, GLYCOLATE OXIDASE (GOX), GDC-H, GLYCERATE 3-KINASE (GLYK), PHOSPHOGLYCOLATE PHOSPHATASE (PGP) were significantly upregulated when compared with *T. diffusa* (C_3_). Only one SHMT isoform (Bv6_143730_mggd.t1) was significantly upregulated in *B. prostrata* (C_4_) when compared with *Sed. sedoides* (C_2_) (Appendix A). In *T. diffusa* (C_3_), only one gene GDC-P was significantly upregulated as compared with *Sed. sedoides* (C_2_) (Appendix A). All photorespiratory genes except GLYK significantly expressed in *Sed. sedoides* (C_2_), compared with *B. prostrata* (C_4_), were significantly upregulated in *T. diffusa* (C_3_) when compared with *B. prostrata* (C_4_).

### 2.6. Regulatory Elements in C_3_, C_2_, and C_4_ Species of the Amaranthaceae/Chenopodiaceae Alliance

To identify regulatory genes putatively involved in the formation/regulation of C_2_ and C_4_ photosynthesis, expression patterns of TFs from The Plant Transcription Factor Database v.5.0 (PlantTFDB; [30,31]) using 1163 annotated TFs from Beta vulgaris (version ‘BeetSet-2′, [32]) were investigated. From the whole set of TFs included in PlantTFDB, 824 orthologous TFs were found, of which 494 TFs had orthologs in all eight species (three species from *Camphorosmeae*: *B. prostrata* (C_4_), *Sed. sedoides* (C_2_), *T. diffusa* (C_3_); five species from *Salsoleae*: *H. soparia* (C_4_), *Sal. divaricata* Pop-184 (C_2_), *Sal. divaricata* Pop-198 (C_2_), *Sal. oppositifolia* (C_4_), *Sal. soda* (C_3_/C_4_), *Sal. webbii* (C_3_)) and were further analysed using Clust. Based on the expression pattern, from the initial 494 TFs, 431 TFs were grouped into 11 clusters with between 22 and 71 genes per cluster (Cluster-C0 to Cluster-10; Figure 3, Appendix A). Nine of the 11 clusters were of particular interest, because these clusters included TFs that were highly expressed in each photosynthetic type when compared with others (Figure 3). Cluster-C4, Cluster-C5, and Cluster-C6 contained TFs that were highly expressed in C_4_ species as compared with C_3_ and C_2_ species, whereas Cluster-C0, Cluster-C9, and Cluster-C10 comprised TFs that were highly expressed in C_3_ species when compared with C_2_ and C_4_ species. Finally, Cluster-C1, Cluster-C2, Cluster-C3 encompassed TFs that were highly expressed in C_2_ species as compared with C_3_ and C_4_ species.

Cluster-C4, Cluster-C5, and Cluster-C6 consisted of 71, 33 and 35 TFs, respectively, from 41 different TF families (Figure 3, Appendix A). Four TFs, all part of Cluster-C4, were present in all eight species, with the transcripts significantly (adjusted *p*-value ≤ 0.01) more abundant in all C_4_ species compared with the two C_3_ species (Table 3). These TFs comprised BBX15 (CO-like family, AT1G25440.1), SHR (TF family GRAS), SCZ (TF family HSF), and LBD41 (TF family LBD). Cluster-C0, Cluster-C9, and Cluster-C10, respectively, comprised 57, 35, and 34 TFs (Figure 3, Appendix A). Here, two TFs of Cluster-10 (HSF, and NAC) and one of Cluster-9 (HD-ZIP) were significantly abundant in the studied C_3_ species when compared with C_4_ species (Table 4).

Cluster-C1, Cluster-C2, and Cluster-C3 included 22, 32, and 38 TFs, respectively, of which one TF of Cluster-2 bHLH 106 (TF family bHLH) was significantly higher in the C_2_ species when compared with the C_3_ species and C_4_ species (Table 5, Appendix A). To assess the integration of C_3_ and C_4_ pathways into the intermediate C_2_ pathway at the regulation level and vice versa, specific TFs of C_4_ (Cluster-C4, Cluster-C5 and Cluster-C6) and C_3_ (Cluster-C0, Cluster-C9, and Cluster-C10) pathways were assessed in the following comparisons: C_2_ species vs. C_3_ species and C_2_ species vs. C_4_ species. Then, specific TFs of the C_2_ pathway (Cluster-C1, Cluster-C2 and Cluster-C3) were estimated in the pairwise comparison of C_3_ species vs. C_4_ species. Among the four TFs common to all C_4_ species, only one TF (BBX15, TF family CO-like) was significantly upregulated in C_2_ species when compared with C_3_ species (Appendix A). Conversely, no TF of C_3_ species was highly expressed in C_2_ species when compared with C_4_ species.

## 3. Discussion

### 3.1. Transcriptome Analysis in Camphorosmeae

Gene expression analysis predominantly paved the way to understand the difference between derived photosynthetic types (C_2_, C_4_) and the ancestral C_3_ photosynthesis [24,34,35]. Much progress in understanding C_4_ and C_2_ photosynthesis was achieved by comparing differentially expressed genes of closely related species in the genus *Flaveria* (*Asteraceae*) considered as a model organism to study the evolution of C_4_ photosynthesis [6,7,34,35,36,37]. The goosefoot family (*Chenopodiaceae*) has a large number of C_2_ and C_4_ species that differ anatomically and ecologically from *Flaveria*. This family therefore represents a good supplementary alternative to decipher the convergent evolution of C_4_ photosynthesis. With PCA based on gene expression, it was possible to clearly distinguish between *T. diffusa* (C_3_), representing the ancestral condition, and *Sed. sedoides* (C_2_) and *B. prostrata* (C_4_), representing derived conditions. The physiologically C_3_-C_4_ intermediate *Sed. sedoides* (C_2_) was positioned in a triangle with *T. diffusa* (C_3_) and *B. prostrata* (C_4_) in terms of transcript variation. This result was similar to what was found in *Salsoleae* [24]. The first three components explained about 76% of the total variation, which was slightly higher than the 73% reported in *Salsoleae* [24]. Similar to *Salsoleae*, in *Camphorosmeae*, the three different photosynthesis types predominantly structure the gene expression pattern in assimilating tissue. Indeed, species with C_3_, C_2_, and C_4_ photosynthesis differ in leaf anatomical structure. *T. diffusa* (C_3_) exhibits the *Neokochia* type characterised by an undifferentiated chlorenchyma of several layers. *Sed. sedoides* (C_2_) has the *Sedobassia* type consisting of kranz-like cells near peripheral vascular bundles. *B. prostrata* (C_4_) depicts the *Bassia prostrata* type with the chlorenchyma differentiated in an outer mesophyll and inner kranz-layer [20]. In contrast, the first three PCA components in a comparable study of *Flaveria* explained only 27% [35]. This difference could be due to the younger evolutionary age or other confounding factors affecting gene expression in the *Flaveria* study as suggested by Lauterbach et al. [24].

### 3.2. C_4_ Key Enzymes in C_4_ and C_2_ Camphorosmeae Species

Analyses of differential gene expression between C_3_ and C_4_ species of *Camphorosmeae* showed that core C_4_ cycle proteins were highly abundant in *B. prostrata* (C_4_). Similar results were found in *Cleome* [38], *Flaveria* [34,35] and *Salsoleae* [24]. Traditionally, three biochemical subtypes of C_4_ photosynthesis are classified according to the predominant type of decarboxylation releasing CO_2_ around RUBisCo in the BSCs: NAD-ME, NADP-ME, and PEP-CK. However, PEPCK should be considered as a supplemental subtype to either NAD-ME or NADP-ME [39]. Significant expression of NADP-ME indicates that *B. prostrata* (C_4_) uses a NADP-ME type C_4_ cycle. Asparagine synthetase (ASN) and NHD were found significantly expressed and upregulated only in *B. prostrata* (C_4_) as compared with *T. diffusa* (C_3_) and *Sed. sedoides* (C_2_). ASN was reported upregulated in C_4_ species *Gynandropsis gynandra* when compared with closely related C_3_ species *Tarenaya hassleriana* (*Cleomaceae*), as well as in C_4_ leaves of *Sal. soda* when compared with its C_3_ cotyledones [23]. On the other hand, NHD was found upregulated in C_4_ species compared with C_3_ and C_2_ species of *Flaveria* [35]. Moreover, the top three highly expressed C_4_ enzymes in *B. prostrata* (C_4_) as compared with *T. diffusa* (C_3_) were Ala-AT, PPDK, and BASS2. ASN is involved in ammonium metabolism and asparagine in nitrogen transport [24]. Achievement of the C_4_ cycle requires the transport of pyruvate to the mesophyll cell (MC) for regeneration of PEP. While Ala-AT plays an important role in pyruvate generation, PPDK intervenes in the regeneration of PEP. Pyruvate transport is mediated by the BASS2/NHD transport system [40]. Taken all together, this indicates not only a possible functional connection between nitrogen metabolism and the switch from C3 to C4 pathway as suggested by Lauterbach et al. [24] and Mallmann et al. [35], but also the capacity to shuttle pyruvate from the BS plastid. In this regard, the pyruvate shuttle ensures the regeneration of the CO_2_ acceptor (PEP), and therefore maintains the C4 pathway.

Eleven C_4_-related genes were found significantly upregulated in *Sed. sedoides* (C_2_) compared with *T. diffusa* (C_3_), including, for example, PEPC, NADP-ME, PPdK, and PHT4. Upregulation of C_4_ typical enzymes such as PEPC, NADP-ME, PPdK, PPT was also reported in the C_2_ species when compared with the C_3_ species in studies of *Flaveria* and *Salsoleae* [24,35]. This result suggests that genes associated with the C_4_ cycle are present in *Sed. sedoides* (C_2_) and play an important role in C_2_ metabolism. A DIT isoform (Bv4_072630_xjai.t1), NADP-ME and Ala-AT were the three most upregulated C_4_ enzymes in *Sed. sedoides* (C_2_) as compared with *T. diffusa* (C_3_). Moreover, we found two transporters (TPT and DIT) upregulated in *Sed. sedoides* (C_2_) when compared with *B. prostrata* (C_4_). These transporters were found highly expressed in some C_2_ species when compared with C_4_ species in *Flaveria* [35]. DIT is a putative plastidial dicarboxylate transporter and TPT is the chloroplast envelope triose-phosphate/phosphate translocator (TPT) [41]. Based on simulated data, it was shown that a high TPT capacity is required to obtain high assimilation rates and to decrease the CO_2_ leakage from BSCs to MCs [39]. The most likely reason for upregulation of these genes is their involvement in decreasing the CO_2_ leakage from the Kranz-like cells back to the MCs due to the presence of RuBisCo, which is not the case for C_4_ plants. This explains the low CO_2_ compensation observed in C_2_ species [42]. Thus, C2 plants upregulate a distinct set of C_4_ enzymes to handle constraints related to the C_2_ pathway and not an entirely congruent set. This does not support their interpretation as an intermediate state towards C4 photosynthesis, but is more in line with their interpretation as an independent evolutionarily stable state ([43] and refs. therein).

### 3.3. C_4_ Key Enzymes in C_4_ and C_2_ Camphorosmeae Species

Transcripts associated with photorespiration were about twice as abundant in C_3_-C_4_ intermediate (*Sed. sedoides*) and *T. diffusa* (C_3_) compared with *B. prostrata* (C_4_). Likewise, we found key photorespiration enzymes were differentially expressed and upregulated in the C_2_ species (*Sed. sedoides*) when compared with *T. diffusa* (C_3_) and *B. prostrata* (C_4_). This corroborates the expression patterns reported in the C_2_ species of *Flaveria* [34,35] and *Salsoleae* [24], implying a successful integration of C_2_ photosynthesis in *Sed. sedoides* (C_2_). Our transcript data showed that GDC-P and a SHMT isoform (Bv6_143730_mggd.t1) were downregulated in *Sed. sedoides* (C_2_) when compared with *T. diffusa* (C_3_) and *B. prostrata* (C_4_), respectively. Similar results were obtained in the C_2_ species of the genus *Flaveria* [35]. Schulze et al. [36] showed that downregulation of GDC-P was closely linked to the establishment of the C_2_ pathway in *Flaveria*. Since GDC-P and a SHMT isoform are known to be involved in glycine decarboxylation, their downregulation in *Sed. sedoides* (C_2_) might have similar consequences. It is worth noticing that the number of significantly upregulated photorespiratory genes in *Sed. sedoides* (C_2_) was equal to *T. diffusa* (C_3_) when compared with *B. prostrata* (C_4_).

A significant reduction in almost all photorespiratory genes was observed in *B. prostrata* (C_4_). All photorespiratory genes except GLYK were downregulated in *B. prostrata* (C_4_) as compared with *T. diffusa* (C_3_). Mallmann et al. [35] reported significant downregulation of all photorespiratory genes in C_4_
*Flaveria* except the transport proteins DIT1 and DIT2 and one isoform of GLDH. On the other hand, GLYK was expressed in the M of C_4_
*Sorghum bicolor* [44]. GLYK catalyses the regeneration of 3-phosphoglycerate (3-PG). The localisation of GLYK within the leaf cells of *B. prostrata* (C_4_) could clarify its high expression and role.

### 3.4. Regulation of C_3_, C_2_ and C_4_ Photosynthesis in Amaranthaceae/Chenopodiaceae

Transcription factors are proteins that bind to the DNA promoter or enhancer regions of specific genes and regulate their expression. They have a crucial role on plant growth, development and adaptation under various stress conditions, and therefore are excellent candidates for modifying complex traits in plants [45]. C_3_, C_2_ and C_4_ species of *Salsoleae* and *Camphorosmeae* are widely spread in desert, semi-desert, saline, and arid regions [18,19]. In former *Chenopodiaceae*, C_4_ photosynthesis evolved as an adaptation to hot, dry, or saline areas from the C_3_ ancestor which was already preadapted to grow in these harsh environments [15]. We focused on TFs that were differentially expressed between C_3_, C_2_, and C_4_ species/states irrespective of the lineage, to further reduce the amount of differentially expressed TFs to a small subset of actually C_4_-, C_2_-, and C_3_-related changes. Indeed, a small number of TFs were found differentially expressed between C_3_, C_2_, and C_4_ species/states.

Cluster analysis showed that BBX15, SHR, SCZ, and LBD41 were co-regulated and significantly more abundant in all C_4_ species irrespective of the lineage when compared with C_3_ species. The families to which these TF families belong play an important role in regulatory networks controlling plant growth and development, and plant adaptive responses to various environmental stress conditions [46,47,48,49,50]. Except the LBD TF family, the SHR, HSF, and CO-like families have been shown to be involved in the development of C4 Kranz anatomy in Zea mays L. and potentially involved in the establishment of C_4_ M and Kranz cell identities [51,52,53,54,55]. However, members of the LBD TF family are key regulators of plant organ development, leaf development, pollen development, plant regeneration, stress response, and anthocyanin and nitrogen metabolisms [50,56]. Since mRNA of all the four TFs was highly abundant in C_4_ species and co-regulated, our data suggest a critical role of these TFs in the development of any C_4_ Kranz anatomy in the *Amaranthaceae*/*Chenopodiaceae* complex.

We found that C_3_ species enhanced different TFs compared with C_4_ species. Three TFs (ATHB13, HD-ZIP family), HSFA6B (HSF TF family), and NAC083 (NAC TF family), in which two TFs (HSF6B and NAC083) are co-regulated, were significantly higher in all C_3_ species when compared with C_4_ species. As C_4_ TFs, they are involved in plant growth, development, and stress tolerance. The NAC TF family was shown to contribute to root and shoot apical meristems formation in Arabidopsis [57,58], organogenesis [59], salt and drought tolerance in Arabidopsis [60], leaf senescence in tobacco [61], and secondary cell wall formation in cotton [62]. The HD-Zip TF family was reported to regulate plant growth adaptation to abiotic stress such as salt and drought in apple and Arabidopsis [63,64]. Interestingly, HD-ZIP, HSF, and NAC TF families were suggested to control the C_4_ photosynthesis in maize and rice [55,65]. However, in these studies, these TF families were detected using development gradient transcriptome comparison only on C_4_ maize and rice plants. This may imply higher activity of these TF families in C_3_ species. Nevertheless, different transcripts of these TFs families were involved when compared with the present study. Thus, a significant expression of these TFs in C_3_ species could indicate a potential function of these TFs in the C_3_ pathway.

In C_2_ species, one transcript of the BASIC HELIX-LOOP-HELIX (bHLH106) protein from the bHLH TF family was found to be upregulated compared with C_4_ and C_3_ species. Two TFs of the bHLH TF family were shown to regulate a C_4_ photosynthesis gene in maize [66]. This upregulation of bHLH106 in all C_2_ species may suggest its possible role in the development and establishment of the C_2_ photosynthesis specificities relative to other photosynthesis types. Interestingly, one C_4_-specific TF (BBX15, TF family CO-like) was significantly higher in C_2_ species when compared with C_3_ species. Thus, this TF could be responsible for similarities of C_4_ photosynthesis found in C_2_ species such as the Kranz-like anatomy. Surprisingly, no C_3_-specific TF was significantly expressed in C_2_ species when compared with C_4_ species. This indicates that C_2_ and C_4_ photosynthesis represent more derived types of photosynthesis compared with C_3_ photosynthesis. Nonetheless, this seems to be inconsistent with the current model of C_4_ evolution which relies heavily on the interpretation of the physiological intermediacy of C_2_ photosynthesis as an evolutionary stepping stone to C_4_ [8]. One would expect C_3_-specific TFs to be higher in C_2_ species when compared with C_4_ species if the C_2_ photosynthetic type represents an intermediate step along the evolution of C_3_-to-C_4_ photosynthesis as revealed by differential expression analysis of core photorespiratory genes in C_2_ and C_3_ species of *Camphorosmeae* (this study) and *Salsoleae* [24].

Taking the results of this study together, the unique derived TF profile of the C_2_ intermediate species suggests an evolutionarily stable state in its own right. Similarities with C_4_ relatives might result from a hybrid origin involving C_3_ and C_4_ parental lineages, parallel recruitment of a number of TFs in C_4_ and C_2_ lineages or common ancestry, and later divergent evolution. The position of *Sedobassia* as sister to *Bassia* (all C_4_) allows all of these three scenarios [16]. For C_2_ species in *Salsoleae*, however, phylogenomic evidence points to a hybrid origin of the *Sal. divaricata* agg. (Tefarikis et al., in prep.). Further phylogenomic analyses are needed to discern if an early hybridisation event of a C_4_ (or ancestral preadapted C_4_) lineage and a C_3_ lineage led to the origin of the *Sedobassia* lineage which then evolved towards stable C_2_ photosynthesis.

## 4. Materials and Methods

### 4.1. Plant Material

Plants of three *Camphorosmeae* species (*Bassia prostrata* (L.) Beck (C_4_), *Sedobassia sedoides* (Schrad.) Freitag and G. Kadereit (C_2_), and *Threlkeldia diffusa* R.Br. (C_3_) (Figure 4, for voucher information see Appendix A) were grown from seeds in custom mixed potting soil in a glasshouse at the Botanic Garden, Johannes Gutenberg University Mainz, Germany at a minimum temperature of 18 °C in the night. Daytime temperatures varied from 25 to 35 °C in the summer and from 20 to 25 °C in the winter. Plants were watered once a week in the winter and twice a week in the summer and kept at 16 h light/ 10 h dark with natural light and an additional light intensity of ca. 300 µmol m−2 s−1. Leaf samples of the three species were harvested between 16th April and 16th May 2014 between 10:30 a.m. and 13:00 p.m., immediately frozen in liquid nitrogen, and stored at −80 °C for RNA extraction.

### 4.2. RNA Isolation and Sequencing

Total RNA extraction, library preparation, and mRNA sequencing were performed as described by Lauterbach et al. [23,24]. Total RNA was extracted from 16–55 mg leaf tissue of *B. prostrata*, *Sed. sedoides*, and *T. diffusa*. Sequencing of 101 bp single-end reads was performed on an Illumina HiSeq2000 platform. For each species, three individuals were sequenced (i.e., biological triplicates). Sequencing reads of these three species are available under study accession PRJEB36559.

### 4.3. Data Access

RNA-Seq data of the five *Salsoleae* species were retrieved from Lauterbach et al. (2017 a, b; study accession numbers PRJNA321979 and PRJEB22023) (Figure 4, Appendix A). These data comprise: cotyledons, and first and second leaf pair of *Salsola soda* (C_3_/C_4_), cotyledons and leaves of the *Salsola divaricata* population 184 (C_2_, Pop-184), *Salsola divaricata* population 198 (C_2_, Pop-198), and *Salsola oppositifolia* (C_4_); leaves of *Salsola webbii* (C_3_); and assimilating shoots of *Hammada scoparia* (C_4_). For all of these samples, triplicates per species and organ were available [23,24].

### 4.4. RNA-Seq Data Processing

Single-end sequencing reads were checked for quality using the FASTQC tool (www.bioinformatics.babraham.ac.uk/projects/fastqc/, accessed on 15 March 2021), and filtered and trimmed using Trimmomatic v.0.38 [67]. For each species, de novo assembly was conducted using quality-filtered reads of all replicates of leaves and, where present, cotyledons of the respective species with default parameters in Trinity v.2.1.1 [68]. Quality of assemblies were assessed with BUSCO v.3.0 (Benchmarking Universal Single-Copy Orthologs, [69]) using the ‘Eudicotyledons odb10′ dataset [70]. Number of contigs of de novo assemblies were reduced by clustering via CD-HIT-EST v.4.7 [71,72] and only contigs with an open reading frame were included in the downstream analysis, which was conducted with TransDecoder v.5.3.0 (github.com/TransDecoder/TransDecoder accessed on 29 February 2020) followed by another round of CD-HIT-EST. Orthology assignment between the nine de novo assemblies was carried out by conditional reciprocal best (crb) BLAST v.0.6.9 [26] run locally using protein-coding sequences of Beta vulgaris (version ‘BeetSet-2′, [32]) as a reference. Only contigs were included in downstream analyses that had ortholog assignments between at least six of the eight species. Besides ‘BeetSet-2′ from *Beta vulgaris*, contigs were annotated using *Arabidopsis* (TAIR10). Reads of each of the replicates were separately mapped against these reduced data sets via bowtie2 v.2.3.4.1 [73]. Re-formatting and final extraction of read counts (excluding supplementary alignments) were carried out in Samtools v.1.3 [74].

### 4.5. Differential Gene Expression Analysis

Read counts were normalised into transcripts per million (TPM) and used for differential gene expression analysis. Here, pairwise comparison between all eight species was statistically evaluated using edgeR [75] in R (R Core Team, 2018). Hierarchical clustering using Pearson’s correlation and principal component analysis of log_2_ transformed read counts were carried out with Multiexperiment Viewer (MeV) v.4.9 (http://mev.tm4.org/ accessed on 5 February 2020). Co-expressed gene clusters of (1) all expressed transcripts and (2) transcripts annotated as transcription factors were carried out with Clust v.1.10.7 [33]. Pathways were defined in MapMan4 categories [29] with the additional category ‘C_4′_. To identify TFs putatively involved in the formation/regulation of C_2_ and C_4_ photosynthesis, the 1163 annotated TFs from Beta vulgaris (version ‘BeetSet-2′, [32]) from The Plant Transcription Factor Database v.5.0 (PlantTFDB; [30,31]) were used. Here, two different datasets were combined (1: leaf transcriptome data of the three *Camphorosmeae* species *T. diffusa* (C_3_), *Sed. sedoides* (C_2_), and *B. prostrata* (C_4_); 2: leaf transcriptome data of the five *Salsoleae* species *Sal. webbii* (C_3_), *Sal. divaricata* Pop-184 (C_2_), *Sal. divaricata* Pop-198 (C_2_), *H. scoparia* (C_4_), *Sal. oppositifolia* (C_4_), and *Sal. soda* (C_4_); for study accession numbers see above) and transcribed TFs grouped using Clust v.1.10.7 [33] and grouping all samples based on the photosynthetic type into the three conditions C_3_, C_2_, and C_4_. VENNY v. 2.1 (https://bioinfogp.cnb.csic.es/tools/venny/ accessed on 12 July 2021) were deployed to find intersected TFs across all pairwise comparisons.

## 5. Conclusions

The transcriptome data of the *Chenopodiaceae* family provided new insight into C_4_ evolution. Proteins encoding for C_4_ transporters (DIT and TPT) were found significantly upregulated in *Sed. sedoides* (C_2_) when compared with *B. prostrata* (C_4_). Upregulation of those transporters reduces CO_2_ leakage from BSCs to MC, which could otherwise be detrimental to C_2_ photosynthesis due to the presence of RuBisCo in the MC. This suggests evolution of a stable C_2_ photosynthesis independent of C_4_ photosynthesis. Combined analysis of TFs of the sister lineages provides further support of this result. Indeed, while one C_4_-specific TF (BBX15) was significantly higher in C_2_ species when compared with C_3_ species, no C_3_-specific TFs were higher in C_2_ species compared with C_4_ species. Finally, apart from well-known TFs involved in the development of C_4_ Kranz anatomy such as SHR, BBX15, SCZ, and LBD41 may also be associated with its development and physiology. Furthermore, bHLH106 could be related to specific C_2_ anatomy and BBX15 to a characteristic C_4_-like expression pattern found in species with C_2_ photosynthesis. This study sheds light on the differentiated regulation and evolution of transcription factors in C_2_ and C_4_ photosynthesis.

## Figures and Tables

**Figure 1 ijms-22-12120-f001:**
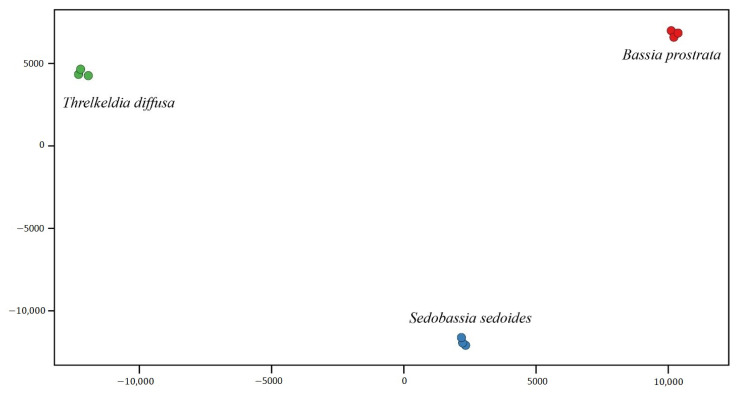
Principal component analysis of log_2_-transformed reads. The first (*x*-axis) and the second (*y*-axis) components are shown, which explain 51.47% and 42.51% of the total variation, respectively.

**Figure 2 ijms-22-12120-f002:**
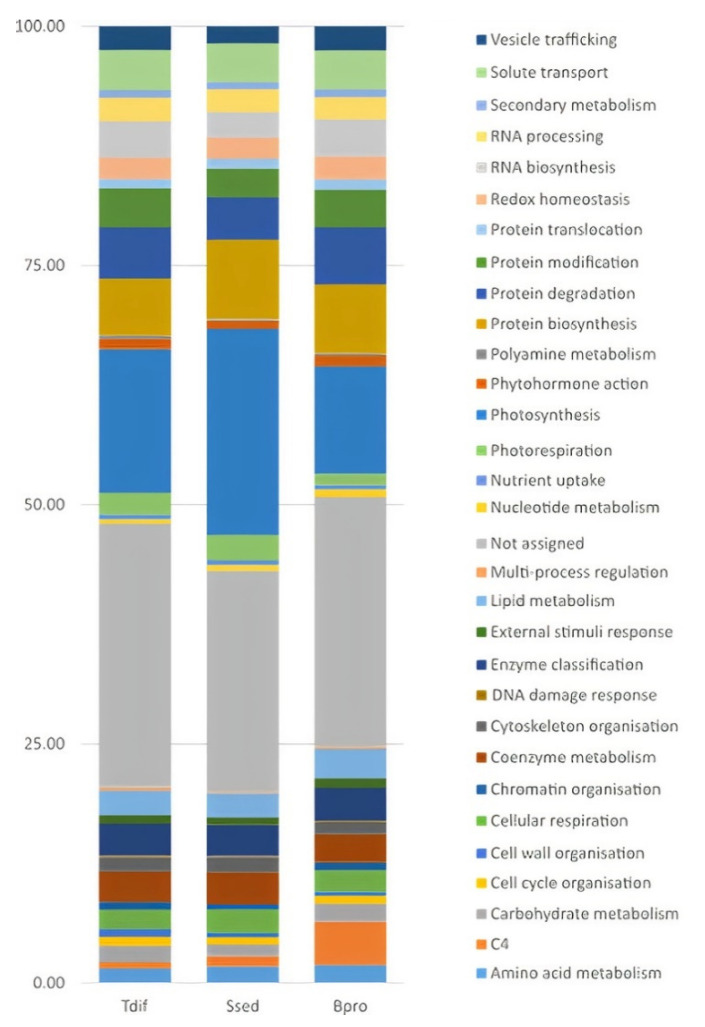
Distribution of transcriptional investment defined as the percentage of all transcripts belonging to a particular MapMan4 category [29] and the additional categories ‘C_4′_.

**Figure 3 ijms-22-12120-f003:**
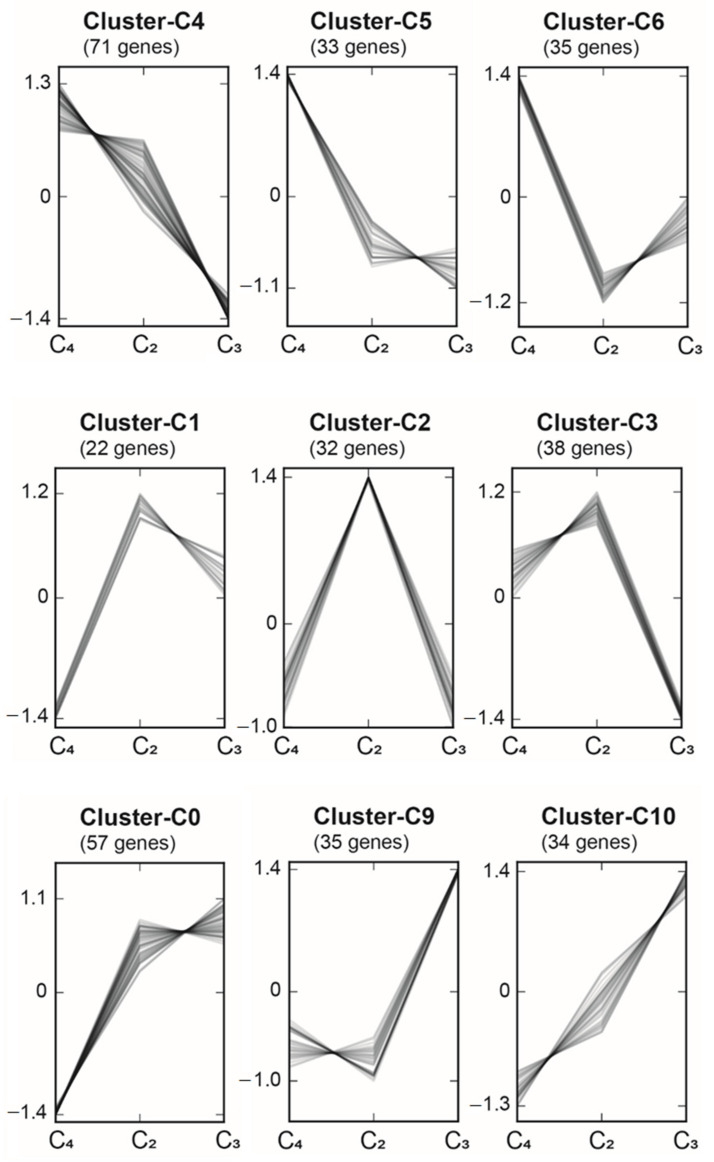
Co-expressed gene clusters were generated using Clust v.1.10.7 [33]. The eight species were grouped into three different conditions ‘C_4_ species’, ‘C_2_ species’, and ‘C_3_ species’. The 11 clusters contained between 22 and 71 genes.

**Figure 4 ijms-22-12120-f004:**
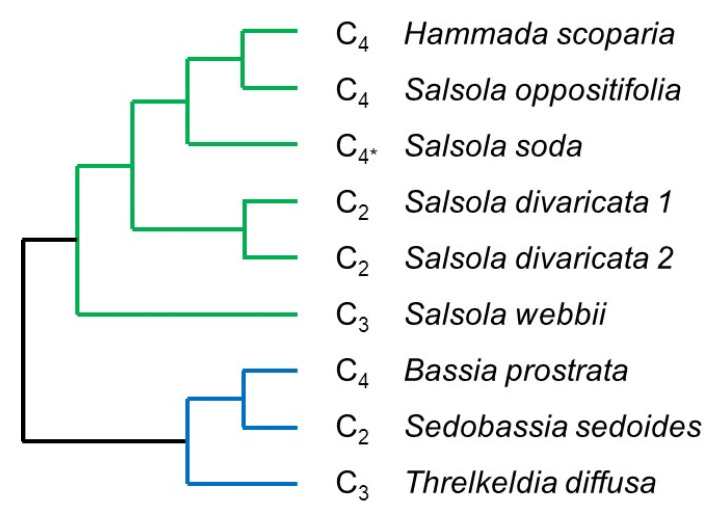
Phylogenetic relationships between species of the current study. The photosynthetic type is indicated. C_4_*, species with C_4_ photosynthesis in leaves/assimilating shoots but C_3_ in cotyledons. Green colour represents *Salsoleae*; blue represents *Camphorosmeae*.

**Table 1 ijms-22-12120-t001:** Differential expression of C_4_-related enzymes in leaves of *Camphorosmeae* species. *T. diffusa* (C_3_), *Sed. sedoides* (C_2_), *B. prostrata* (C_4_).

		C_2 vs._ C_4_*		C_3_ vs. C_4_*		C_3_ vs. C_2_*	
Transcripts	Genes	Log_2_FC	Padj	Log_2_FC	P	Log_2_FC	Padj
Bv_006710_gkqg.t1	MDH	1.29	1.01 × 10^−10^	2.84	9.92 × 10^−42^	1.55	1.74 × 10^−14^
Bv1_004490_tyfq.t1	PHT4	1.57	1.56 × 10^−11^	2.89	1.72 × 10^−31^	1.32	7.56 × 10^−08^
Bv1_013550_fjqs.t1	PPdK	3.37	2.32 × 10^−54^	4.63	2.78 × 10^−90^	1.26	8.92 × 10^−10^
Bv2_031080_twkf.t1	AspAt	1.32	2.84 × 10^−06^	2.15	1.44 × 10^−13^	0.82	0.003667
Bv3_049110_qgnh.t1	PHT1	3.16	8.05 × 10^−08^	-	-	-	-
Bv4_072630_xjai.t1	DIT	-	-	1.30	4.88 × 10^−06^	2.61	3.46 × 10^−19^
Bv5_117240_yhsk.t1	PPT	1.94	7.13 × 10^−13^	3.61	5.61 × 10^−35^	1.67	2.18 × 10^−09^
Bv6_135140_uyxu.t1	Asn Synthetase	2.56	5.60 × 10^−07^	2.42	1.93 × 10^−06^	-	-
Bv6_148840_uffy.t1	CA	3.80	6.66 × 10^−38^	3.81	4.08 × 10^−38^	-	-
Bv7_169130_kwer.t1	AlaAT	2.61	4.89 × 10^−17^	5.17	4.74 × 10^−48^	2.56	2.34 × 10^−15^
Bv8_182550_kstq.t1	TPT	-	-	1.48	1.20 × 10^−10^	2.29	2.71 × 10^−22^
Bv8_194450_rkme.t1	CA	-	-	-	-	1.48	0.000263
Bv8_195530_sxjq.t1	BASS2	3.51	5.00 × 10^−68^	3.97	6.26 × 10^−83^	0.46	0.018625
Bv8_200290_ujgk.t1	TPT			3.25	1.25 × 10^−05^	1.97	0.005513
Bv9_209750_xeaz.t1	PEPC	2.95	3.29 × 10^−17^	4.12	6.98 × 10^−29^	1.18	0.000429
Bv9_215520_prze.t1	DIT	1.48	1.23 × 10^−08^	2.63	5.15 × 10^−22^	1.16	2.27 × 10^−05^
Bv9_224000_xpgi.t1	NHD	2.48	1.17 × 10^−21^	2.68	7.04 × 10^−25^	-	-
Bv9_224840_zmjw.t1	NADP-ME	1.29	8.44 × 10^−08^	3.86	9.54 × 10^−47^	2.56	1.34 × 10^−23^

- not significantly expressed Padj > 0.05.

**Table 2 ijms-22-12120-t002:** Differential expression of photorespiratory transcripts in leaves between *Camphorosmeae* species. *T. diffusa* (C_3_), *Sed. sedoides* (C_2_), *B. prostrata* (C_4_).

		C_2_* vs. C_4_		C_3_ vs. C_2_*		C_3_* vs. C_4_	
Transcripts	Genes	Log_2_FC	Padj	Log_2_FC	Padj	Log_2_FC	Padj
Bv5_106360_ipey.t1	GDC-H	1.862968	3.37 × 10^−15^	0.990549	1.80 × 10^−05^	0.872432	0.000167
Bv_012000_yknj.t1	GDC-P	1.253616	9.20 × 10^−08^	-	-	2.068131	5.39 × 10^−18^
Bv3_059720_tshd.t1	GDC-L	1.281024	1.13 × 10^−11^	-	-	0.953812	3.95 × 10^−07^
Bv3_065510_eeis.t1	SHMT	-	-	1.133566	0.000228	-	-
Bv4_073470_iswc.t1	AGT/SGT	1.254967	2.18 × 10^−07^	-	-	1.130439	2.83 × 10^−06^
Bv4_074740_miaa.t1	PGP	1.831751	1.99 × 10^−16^	0.505181	0.019216	1.326602	1.77 × 10^−09^
Bv4_094290_jgpp.t1	GOX	1.708751	1.51 × 10^−18^	0.993515	1.94 × 10^−07^	0.715241	0.000175
Bv5_107350_ydma.t1		2.435018	1.25 × 10^−13^	-	-	2.32744	1.09 × 10^−12^
Bv6_127540_qdph.t1	GDC-T	2.464812	2.21 × 10^−23^	1.030248	1.32 × 10^−05^	1.434602	2.66 × 10^−09^
Bv6_148110_nuir.t1	GGT	2.380809	3.21 × 10^−21^	1.223501	4.63 × 10^−07^	1.157324	2.02 × 10^−06^
Bv6_152820_wtfn.t1	SHMT	1.721992	2.64 × 10^−18^	0.734934	0.000137	0.987067	3.75 × 10^−07^
Bv8_184280_guso.t1		0.677745	0.006074	-	-	1.06935	3.85 × 10^−06^
Bv9_213980_zwen.t1	HPR	0.96884	0.000282	-	-	0.531187	0.045313
Bv9_220360_xogt.t2	GLYK	0.689722	0.001207	0.708265	0.000885	-	-

- not significantly expressed Padj > 0.05.

**Table 3 ijms-22-12120-t003:** Differentially expressed C_4_-related TFs in *Amaranthaceae*/*Chenopodiaceae*.

		Cluster-4							
		BBX15 (CO-Like)		SHR (GRAS)		SCZ (HSF)		LBD41 (LBD)	
Lineage	Species (C_3_ vs. C_4_*)	LF_2_C	Padj	LF_2_C	Padj	LF_2_C	Padj	LF_2_C	Padj
*Salsoleae*	*Salweb* vs. *Hsco*	8.8	3.84 × 10^−24^	2.23	1.55 × 10^−08^	5.59	1.73 × 10^−05^	3.71	6.05 × 10^−13^
	*Salweb* vs. *Salopp*	11.86	6.56 × 10^−61^	1.95	2.56 × 10^−06^	5.09	0.000374	4.26	1.64 × 10^−17^
	*Salweb* vs. *Salsod*	11.57	6.71 × 10^−62^	2.63	1.85 × 10^−13^	7.66	4.94 × 10^−20^	2.84	2.06 × 10^−07^
*Camphorosmeae*	*Tdif* vs. *Bpro*	10.99	3.77 × 10^−51^	4.21	9.49 × 10^−12^	3.4	2.71 × 10^−06^	6.71	5.11 × 10^−10^
	*Salweb* vs. *Bpro*	10.82	2.50 × 10^−47^	1.45	0.0011	6.12	1.58 × 10^−07^	2.22	0.00019
*Salso.* × *Camph.*	*Tdif* vs. *Hsco*	8.96	2.75 × 10^−27^	4.99	2.89 × 10^−19^	2.84	0.000272	8.23	1.45 × 10^−21^
	*Tdif* vs. *Salsod*	11.72	1.85 × 10^−63^	5.34	5.60 × 10^−25^	5.25	3.76 × 10^−21^	6.96	1.91 × 10^−12^
	*Tdif* vs. *Salopp*	12.03	6.61 × 10^−65^	4.7	1.56 × 10^−15^	2.44	0.00602	8.68	3.84 × 10^−26^

**Table 4 ijms-22-12120-t004:** Differentially expressed C_3_-related TFs in *Amaranthaceae*/*Chenopodiaceae*.

		Cluster-9		Cluster-10			
		ATHB13 (HD-ZIP)		HSFA6B (HSF)		NAC083 (NAC)	
Lineage	Species (C_4_ vs. C_3_*)	LF_2_C	Padj	LF_2_C	Padj	LF_2_C	Padj
*Salsoleae*	*Hsco* vs. *Salweb*	9.02	5.45 × 10^−24^	1.21	0.00017	0.73	0.00398
	*Salopp* vs. *Salweb*	1.71	2.77 × 10^−05^	1.54	1.32 × 10^−05^	1.42	4.44 × 10^−08^
	*Salsod* vs. *Salweb*	1.14	0.00077	1.43	1.65 × 10^−06^	0.76	0.0004
*Camphorosmeae*	*Bpro* vs. *Tdif*	8.89	2.80 × 10^−23^	1.7	1.08 × 10^−06^	11.74	3.28 × 10^−82^
	*Bpro* vs. *Salweb*	8.99	8.45 × 10^−24^	1.91	5.76 × 10^−08^	11.28	1.31 × 10^−72^
*Salso.* × *Camph.*	*Hsco* vs. *Tdif*	8.92	1.78 × 10^−23^	1	0.00169	1.01	5.11 × 10^−05^
	*Salsod* vs. *Tdif*	0.87	0.00956	1.23	4.09 × 10^−05^	1.04	6.71 × 10^−07^
	Salopp vs. Tdif	1.45	0.00038	1.34	0.00011	1.71	4.00 × 10^−11^

**Table 5 ijms-22-12120-t005:** Differentially expressed C_2_-related TF in *Amaranthaceae*/*Chenopodiaceae*.

		Cluster-2	
		bHLH106 (bHLH)	
**Lineage**	**Species (C_3_ vs. C_2_*)**	**LF_2_C**	**Padj**
*Salsoleae*	*Sweb* vs. *Saldi1*	1.68	8.90 × 10^−13^
	*Sweb* vs. *Saldi2*	1.63	4.07 × 10^−11^
*Camphorosmeae*	*Tdif* vs. *Sedsed*	0.45	0.04616
*Salso.* × *Camph*.	*Tdif* vs. *Sdi1*	1.06	3.33 × 10^−07^
	*Tdif* vs. *Sdi2*	1.01	2.79 × 10^−06^
	*Sweb* vs. *Sedsed*	1.07	4.19 × 10^−05^
	**Species (C_4 vs._ C_2_*)**	**LF_2_C**	**Padj**
*Salsoleae*	*Hsco* vs. *Saldi1*	1.37	1.23 × 10^−10^
	*Hsco* vs. *Saldi 2*	1.32	3.58 × 10^−09^
	*Salopp* vs. *Saldi1*	2.17	7.37 × 10^−19^
	*Salopp* vs. *Saldi2*	2.12	1.03 × 10^−16^
	*Salsod* vs. *Saldi1*	1.21	3.24 × 10^−10^
	*Salsod* vs. *Saldi2*	1.16	7.30 × 10^−09^
*Camphorosmeae*	*Bpro* vs. *Sedsed*	0.56	0.01398
*Salso.* × *Camph.*	*Bpro* vs. *Saldi1*	1.17	2.14 × 10^−08^
	*Bpro* vs. *Saldi2*	1.12	2.21 × 10^−07^
	*Hsco* vs. *Sedsed*	0.76	0.001176
	*Salopp* vs. *Sedsed*	1.56	9.00 × 10^−09^
	*Salsod* vs. *Sedsed*	0.6	0.00483

## Data Availability

Data are available upon request.

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
