# Peer review of "Insights into Regulation of C2 and C4 Photosynthesis in Amaranthaceae/Chenopodiaceae Using RNA-Seq"

_ijms, 2021, doi:10.3390/ijms222212120_

Round 1

Reviewer 1 Report

Review of ijms-1439145

Insights into regulation of C2 and C4 photosynthesis in Amaranthaceae/Chenopodiaceae using RNA-Seq

Christian Siadjeu, Maximilian Lauterbach and Gudrun Kadereit

The authors set out to study the evolutionary relationships between plants using C2, C3 and C4 photosynthesis. In particular, they wished to test the hypothesis that C2 photosynthesis is an intermediate step in the evolution of C4 photosynthesis starting from C3 plants. They therefore performed RNAseq on C2, C3 and C4 representatives of the Camphorosmeae in order to identify genes that were differentially expressed I n these three clades and might therefore be associated with the development of these different photosynthetic pathways. Their data identified genes that were differentially regulated in plants using each of these pathways, and provided support for the hypothesis that C2 photosynthesis is a distinct pathway for reducing carbon loss due to photorespiration rather than an intermediate step in the evolution of C4 photosynthesis.

Overall, their results are interesting. The work seems to have been performed competently with adequate replication, and the analyses seem appropriate. I therefore recommend sharing them with the plant community

To me it seems that their key finding supports the idea that C2 photosynthesis is a separate, evolutionarily stable pathway, rather than an intermediate step in the evolutionary transition from C3 to C4 photosynthesis. Although this idea has been previously proposed, this report provides powerful supporting data from a new group of plants. I would therefore like to see the authors make it easier to understand their data that support this conclusion in their results and discussion sections.

Overall the English is very good, but there are a few mistakes. Most involve singular versus plural, or similar simple mistakes.

For example

Line 40 “ are” should be “is” , line 41 “lineage” should be “lineages” Line 45 “was” should be “were”

Lines 58, 65, 73, 248 “photosynthesis types” should be “photosynthetic types”

Line 30 “less” should be “fewer” to avoid confusion with the idea that it produce compounds of lower toxicity.

Line 33: please rewrite for clarity.

Lines 68-70: please rewrite for clarity and grammatical correctness.

Lines 158-168: please rewrite for clarity and grammatical correctness.

Lines 209-213: please rewrite for clarity and grammatical correctness. This sentence is crucial to the message from your paper, so please make it as easy to understand as possible!

Lines 228-234: please rewrite for clarity and grammatical correctness.

Lines 251-254: please rewrite for clarity and grammatical correctness. I don’t understand “convergent evolution” in this sentence, if C2 is historically regarded as an intermediate step in the evolution of C4 photosynthesis.

Lines 262-266: please rewrite for clarity and grammatical correctness.

Lines 293-294: please rewrite for clarity and grammatical correctness.

Lines 304-307: please rewrite for clarity and grammatical correctness.

Lines 347-348: please rewrite for clarity and grammatical correctness.

Lines 406-409: please rewrite for clarity and grammatical correctness

Author Response

Thanks for considering our work and we are grateful for the constructive comments for revision and improvement. Below we addressed each comment

# Reviewer 1

- Line 40 “ are” should be “is” , line 41 “lineage” should be “lineages” Line 45 “was” should be “were”

revised as requested Lines 40, 42, 44

- Lines 58, 65, 73, 248 “photosynthesis types” should be “photosynthetic types”

revised as requested

- Line 30 “less” should be “fewer” to avoid confusion with the idea that it produce compounds of lower toxicity.

revised as requested: line 30

- Line 33: please rewrite for clarity.

Revised as requested: lines 33-34

- Lines 68-70: please rewrite for clarity and grammatical correctness.

Revised as requested: lines 69-73.

- Lines 158-168: please rewrite for clarity and grammatical correctness.

Revised as requested: lines 162-174.

Lines 209-213: please rewrite for clarity and grammatical correctness. This sentence is crucial to the message from your paper, so please make it as easy to understand as possible!

Revised as requested: lines 214-223.

- Lines 228-234: please rewrite for clarity and grammatical correctness.

I deleted these lines because the information was redundant after rewriting for clarity lines 209-213.

- Lines 251-254: please rewrite for clarity and grammatical correctness. I don’t understand “convergent evolution” in this sentence, if C2 is historically regarded as an intermediate step in the evolution of C4 photosynthesis.

Revised as requested lines 259-266

- Lines 262-266: please rewrite for clarity and grammatical correctness.

Revised as requested: lines 274-279.

- Lines 293-294: please rewrite for clarity and grammatical correctness.

revised as requested: lines 310-311

- Lines 304-307: please rewrite for clarity and grammatical correctness.

Revised as requested: lines 322-324 “

- Lines 347-348: please rewrite for clarity and grammatical correctness.

Revised as requested lines 364-365.

- Lines 406-409: please rewrite for clarity and grammatical correctness

Revised as requested: lines 426-429.

Reviewer 2 Report

The present study deals with the differential nature and regulation of TFs in C2, C3 and C4 photosynthesis species. The study is well-written, well-conducted and presents well-supported conclusions.

Few minor remarks towards imprevent:

(1) line 9: shows in place of show

(2) Line 19 are in place of is

(3) I understand that the focus of the paper is the transition from C3 to C4. What about the CAM plants, and their transition for C3 to CAM state (Fanourakis et al., 2017 Environ Exp Bot 143, 115–124). (see line 67) 

(4) The species are correctly in italics in the first part of the paper, but later on this is neglected (Lines 100, 111, 124, 129, etc.)

(5) Line 77: why give the full word (transcription factors) since the abbreviations (TFs) has been introduced?

(6) line 83: is the term hot appropriate here?

(7) line 94/ 404: is the term sister tribe appropriate here?

(8) lines 416-419: photoperiod, light quality of assimilation light, air temperature, relative air humidity, watering? which leaf was sampled? fully-expanded or growing one? is the time of sampling important for photosynthesis-related TFs? 

(9) for the general audience, what is the importance of the obtained findings? can one employ these for breeding purposes? can one employ them for a better understanding of photosynthesis performance? what are the implications for cultivated (crop) plants? or this is purely for fundamental purposes

(10) what is left to be done? what is expected when comparing the TF profiles with those of CAM plants?

(11) Lines 406: a submitted manuscript cannot be cited

Author Response

Thanks for considering our work and we are very grateful for the constructive comments for revision and improvement. Below we addressed each comment in detail

# Reviewer 2

(1) line 9: shows in place of show

Revised as requested line 9

(2) Line 19 are in place of is

Revised as requested line 19

(3) I understand that the focus of the paper is the transition from C3 to C4. What about the CAM plants, and their transition for C3 to CAM state (Fanourakis et al., 2017 Environ Exp Bot 143, 115–124). (see line 67) 

Thanks for pointing this out. However, as you mentioned, our focus was on the transition C3 to C4. Secondly, the transition C3 to CAM in the paper (Fanourakis et al., 2017) that you shared with us is a state transition (only in leaves) depending on the environment conditions, whereas our transition is a development transition from cotyledons C3 to leaves C4.

(4) The species are correctly in italics in the first part of the paper, but later on this is neglected (Lines 100, 111, 124, 129, etc.)

Revised as requested

(5) Line 77: why give the full word (transcription factors) since the abbreviations (TFs) has been introduced?

Revised as requested line 81

(6) line 83: is the term hot appropriate here?

Yes the term is appropriate to outline how important are transcription factors for the evolution of complex traits in general

(7) line 94/ 404: is the term sister tribe appropriate here?

We replaced sister tribe by sister lineage: line 98. In line 404 there was no tribe in the sentence: line 423

(8) lines 416-419: photoperiod, light quality of assimilation light, air temperature, relative air humidity, watering? which leaf was sampled? fully-expanded or growing one? is the time of sampling important for photosynthesis-related TFs? 

More details regarding photoperiod, temperature , light intensity, watering in greenhouse were added Lines 436-439. Adult and not fully-expanded leaves were sampled. Yes, Sampling time might affect the expression level of photosynthesis related TFs because light intensity affects photosynthesis and does vary within the day with the highest peak intensity between 10am to 2pm.

(9) for the general audience, what is the importance of the obtained findings? can one employ these for breeding purposes? can one employ them for a better understanding of photosynthesis performance? what are the implications for cultivated (crop) plants? or this is purely for fundamental purposes

For the general audience, our findings show that plants with C2 photosynthesis might be hybrids between C3 and C4 plants, which would pave the way for breeding for C4 photosynthesis. Yes, one could use C3 x C4 hybrid for a better understanding of photosynthesis performance. For crop plants, that means breeding for climate resilient traits.

(10) what is left to be done? what is expected when comparing the TF profiles with those of CAM plants?

Further studies are needed to investigate roles of these TFs in other lineages. When comparing with CAM plants, we would expect to observe the similar TF profiles since CAM plants use C4 photosynthesis as well.

(11) Lines 406: a submitted manuscript cannot be cited

Instead of submitted we added in prep line 425